# Numerical Prediction of Erosion of Francis Turbine in Sediment-Laden Flow under Different Heads

Jinliang Wang [1], Xijie Song [2], Hu Wang [1], Ran Tao [3] and Zhengwei Wang [2,*]

1    Power Station Management Bureau, Y.R. Wanjiazhai Water Multi-Purpose Dam Project Co., Ltd., Taiyuan 036412, China; wangjinliangwjz@sohu.com (J.W.)
2    State Key Laboratory of Hydroscience and Engineering, Department of Energy and Power Engineering, Tsinghua University, Beijing 100084, China; songxijie@mail.tsinghua.edu.cn
3    College of Water Resources and Civil Engineering, China Agricultural University, Beijing 100083, China
*    Correspondence: wzw@mail.tsinghua.edu.cn

**Abstract:** Hydropower stations are an important source of clean energy, usually operating in sandy water flow, and the turbine wheels may suffer severe wear and tear. In addition, during the operation of the unit, it is necessary to operate at different water heads according to the actual situation, which will result in varying degrees of wear and tear. In this paper, the Lagrange method is used to study the wear characteristics of a Francis turbine under different water heads. The research object is the water turbine in Wanjiazhai Hydropower Station. Research has shown that wear on the walls of the turbine volute, guide vanes, and runner is inevitable, and the clearance walls are also vulnerable to wear. The difference in the water head mainly affects the movement trajectory and impact speed of particles. The higher the water head, the more severe the wear on the wall surface of the flow passage components. Both the crown and lower ring of the runner are worn. The impact of particles causes wear at this location, and the greater the relative velocity relative to the runner, the more severe the wall wear. This indicates that reasonable head operating conditions can effectively reduce wall wear, which provides guidance for the operation of hydraulic turbines.

**Keywords:** Francis turbine; guide vane opening; sediment erosion; sediment-laden flow; different heads

## 1. Introduction

Sediment erosion damage is a complex physical phenomenon that has always received widespread attention from many scholars and engineering technicians both domestically and internationally due to its destructive nature. Sediment erosion not only depends on the diameter, hardness, shape, and content of sand particles, but is also related to the material characteristics, operating conditions, impact conditions, and other factors of hydraulic turbines [1–3].

At present, research on the sediment erosion resistance of hydraulic turbines mainly includes two aspects: some scholars focus on studying the improvement of metal component materials and surface coatings. Part of the research focuses on optimizing the internal flow field of hydraulic turbines to weaken the interaction between solid particles and metal components, thereby improving the sediment erosion resistance of components [4]. Cheng et al. [5] and Song [6] used numerical simulations to explore the flow characteristics of solid–liquid two-phase flow in the internal sediment erosion of hydraulic turbines, predicting the location and degree of sediment erosion in the flow channel of the water turbine. Matsumura et al. [7] and Koirala et al. [8] studied the effects of sediment particle concentration, size, and shape on erosion and efficiency loss through numerical simulation and found that with the increase in sediment particle concentration and size, erosion and efficiency loss both increase.

The numerical simulation of two-phase flow law is based on the calculation of a clean water flow field, and the development of computational fluid dynamics also provides necessary basic conditions for the numerical simulation of two-phase flow [9]. At present, there are two main numerical methods to deal with two-phase flow: the Euler Euler method and the Euler Glanges method. In this paper, the Lagrangian Particle Tracking Implementation is proposed for the calculation. The basic assumptions of this model are that the particles are spherical and the particle phase has local characteristics, which are obtained by averaging the particle trajectories passing through each specific control body in space [10]. It is impossible to track all particles and ignore the details of the flow field around the particles. The inter phase coupling effect caused by the source term is the difference between two-phase flow simulation and single-phase flow [11]. There are two types of inter phase coupling: one is called unidirectional coupling, which does not consider the influence of particle existence on the fluid when the particle concentration is low, but only considers the influence of the fluid on particle trajectories and parameters. One type is called bidirectional coupling, which considers the interaction between particles and fluid when the particle concentration is high enough [12].

There are two commonly used wall erosion models. One is the Finnie model, whose erosion amount is a function of velocity and impact angle. The other is the Tabakoff erosion model [13], which includes more available solid parameters and has been verified by the test results of Francis turbine components and materials. Zhang et al. [14] analyzed the erosion condition of the turbine by using the Tabakoff erosion model. Therefore, the operation strategy of the power station is optimized. Smirnov et al. [15] analyzed the sediment erosion of a low specific speed Francis turbine by using the Tabakoff erosion model and compared the results with the test, which shows that the degree of erosion is positively related to the size of sediment particles.

In this article, the Francis turbine is taken as the research object to study the sediment erosion problem under the inflow conditions of high sediment concentration water, based firstly on the *N-S* equation and SST $k$-$\omega$ turbulence model, the Tabakoff erosion model for wall surface, and the CFX solver for full channel numerical simulation. Secondly, the location and degree of sediment erosion in the hydraulic turbine is analyzed, as well as the characteristics of sediment erosion inside the flow channel of the hydraulic turbine in order to provide reference for the prediction of erosion in the hydraulic turbine and the renovation design of the power plant hydraulic turbine.

## 2. Numerical Simulation Setup

### *2.1. Mathematical Model*

2.1.1. Governing Equations

In this paper, the Eulerian–Lagrangian method is used to simulate the discrete phase particles in the sediment-laden stream of the water turbine [13–15]. The flow control equation of continuous phase is solved by *N-S* equation.

$$\frac{\partial(\rho u)}{\partial t} + \nabla\cdot(\rho u u) = -\nabla p + \rho v \Delta u - \rho \nabla\cdot\tau + S_t \tag{1}$$

where $u$ is the flow velocity, $t$ is the time, $\rho$ is the fluid density, $p$ is the flow pressure, $v$ is the kinematic viscosity of the fluid, and $S_t$ is the source term. $\tau$ is the Reynolds stress defined as:

$$\tau = \tau^d + \frac{2k}{3}\delta \tag{2}$$

where $\tau^d$ is the deviatoric Reynolds stress, $k$ is the turbulent kinetic energy, and $\delta$ is the Kronecker delta. Based on viscosity ($v_t$) assumption, Equation (2) can be written as:

$$\tau = -2v_t S + \frac{2k}{3}\delta \tag{3}$$

where, $S$ is the train-rate tensor,

$$S = \frac{1}{2}\left(\nabla u + \nabla^T u\right) \tag{4}$$

### 2.1.2. Particle Tracking Model

In this paper, the flow description of the particle phase in the Lagrangian coordinate system is assumed to be spherical in shape, and the control equation follows the generalized Newton's second law [16].

$$m_p \frac{du_p}{dt} = F_D + F_B + F_G + F_V + F_P + F_X \tag{5}$$

where $t$ is time, $m_p$ is particle mass, $u_p$ is particle velocity, $F_D$ is resistance, $F_B$ is Basset force, $F_G$ is gravity, $F_V$ is virtual mass force, $F_P$ is pressure gradient force, and $F_X$ is the sum of other external forces considered.

### 2.1.3. Erosion Model

In this paper, the Tabakoff and Grant erosion model is adopted to predict the internal sediment erosion of water turbine. The formula is as follows:

$$E = f(\gamma)\left(\frac{V_p}{V_1}\right)^2 \cos^2\gamma\left[1 - \left(1 - \frac{V_p}{V_3}sin\gamma\right)^2\right] + \left(\frac{V_p}{V_2}sin\gamma\right)^4 \tag{6}$$

$$f(\gamma) = \left[1 + k_1 k_{12} \sin\left(\gamma\frac{\pi/2}{\gamma_0}\right)\right]^2 \tag{7}$$

Here, $\gamma_0$ is the angle of maximum erosion. $k_1$ to $k_4$, $k_{12}$, and $\gamma_0$ are model constants and depend on the particle/wall material combination.

### 2.2. Geometric Model Set-up

Geometric modeling of the calculation model, as shown in Figure 1. The maximum thickness of the runner blade is 32 mm, the runner diameter is 2.3 m, and the clearance thickness is 1.5 mm. The rotating speed is 100 r/min. A Workbench mesh is used to mesh the calculation model, as shown in Figure 2, and a hexahedral structure mesh is drawn at the gaps.

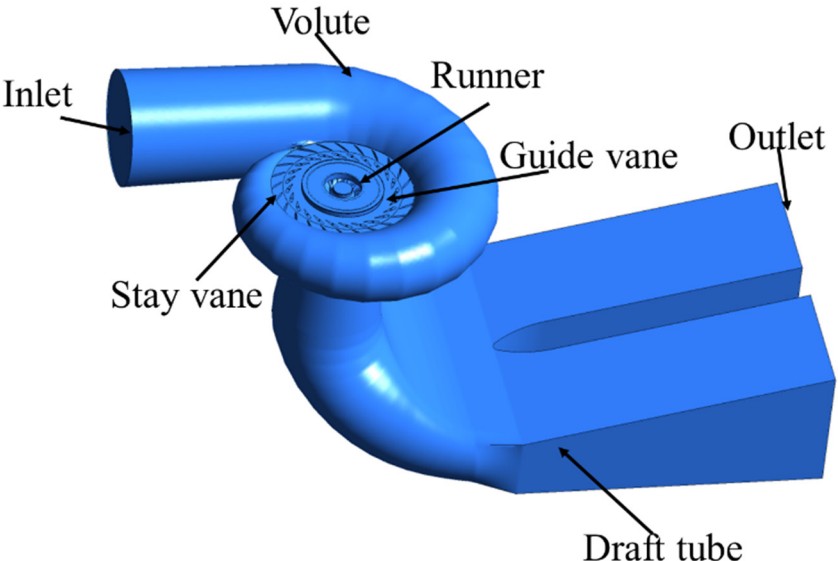

**Figure 1.** Model of numerical simulation.

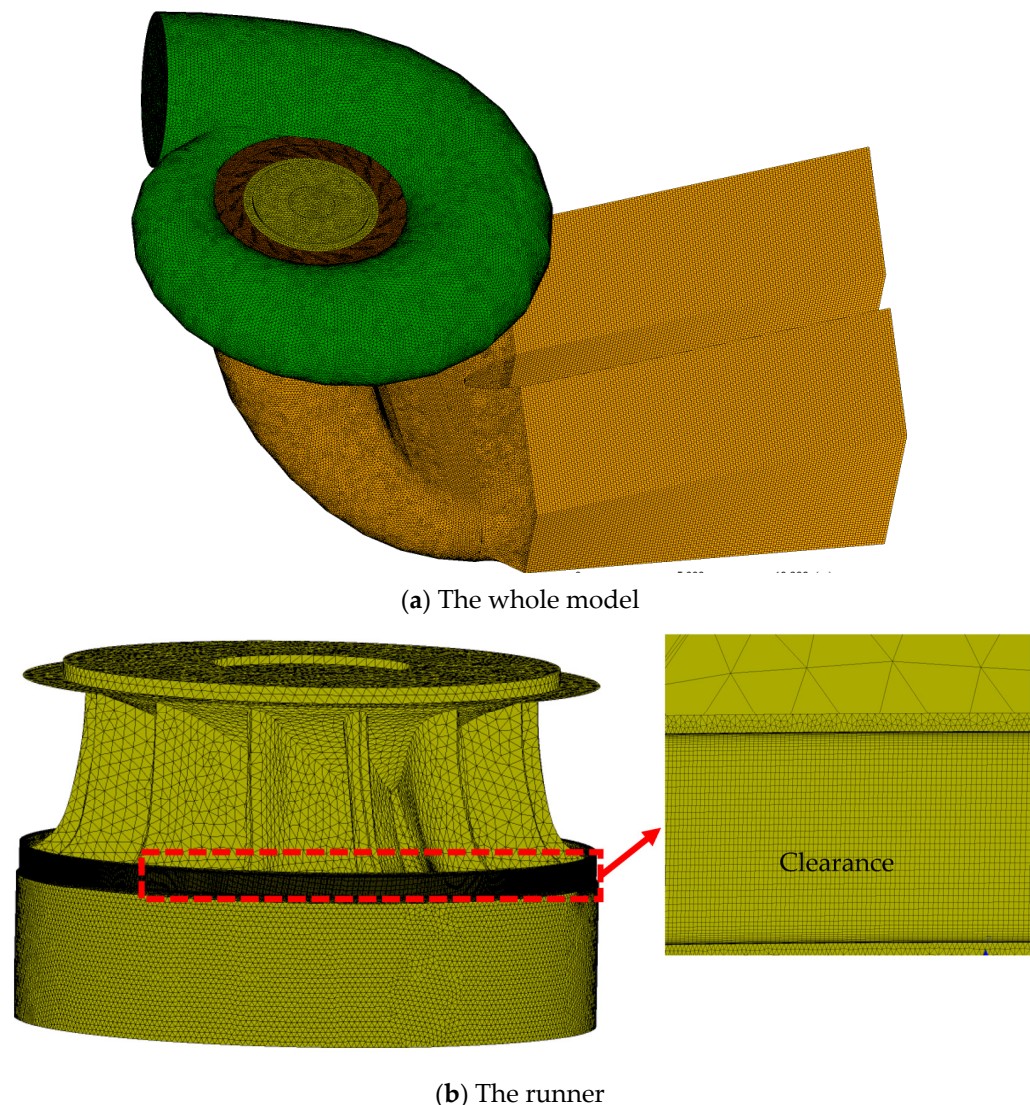

(**a**) The whole model

(**b**) The runner

**Figure 2.** Grid division distribution map.

In numerical simulation, the quality and number of grids have a significant impact on the accuracy and speed of the calculation results. Generally, the larger the number of grids, the higher the quality of the grid. However, the more grids there are, the more computing resources are required. Therefore, in numerical simulation, it is necessary to verify the independence of the grid to select the appropriate number of grids.

Grid independence verification was conducted using the efficiency of the unit and retrograde flow. A total of four calculation schemes with grid numbers of 12 million, 15 million, 18 million, and 21 million were designed, and the results showed that when the number of grids reached 18 million, the operational efficiency of the unit was no longer affected by the number of grids, as shown in Figure 3. Therefore, the calculation model had 18 million grids.

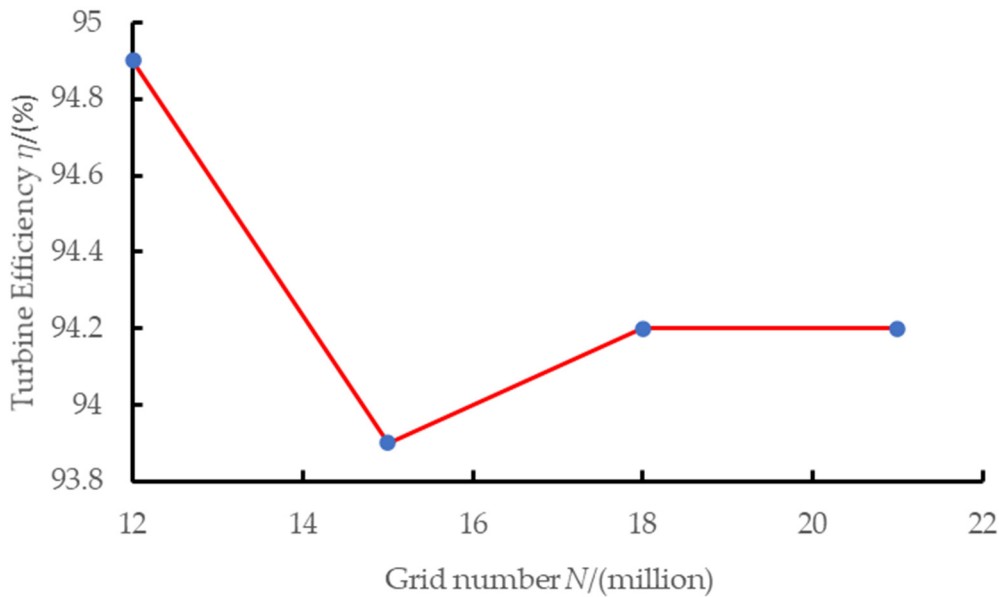

**Figure 3.** Grid independence verification curve.

### 2.3. Parameter Setting in Calculation Model

In this paper, the sediment particles are spherical particles with a particle size of 0.02 mm and a particle density of 2300 kg/m$^3$. The inlet boundary was set as the total pressure, and the outlet boundary was adopted as the static pressure condition related to the water level downstream. The interface between the runner and stationary parts adopts the dynamic static interface. In the calculation scheme, the particle concentration is 15 kg/m$^3$, and the guide vane openings are designed as 24°. The turbulence model adopts the SST $k$-$\omega$ model.

### 2.4. Calculation Scheme

The purpose of this paper is to explore the internal sediment erosion characteristics of the Francis turbine under different water heads and select the turbine of Wanjiazhai Hydropower Station as the research object.

Figure 4 shows the comprehensive characteristic curve of the water turbine of Wanjiazhai hydropower station. $H$ is the water head of the hydraulic turbine. P is the operating power of the hydraulic turbine.

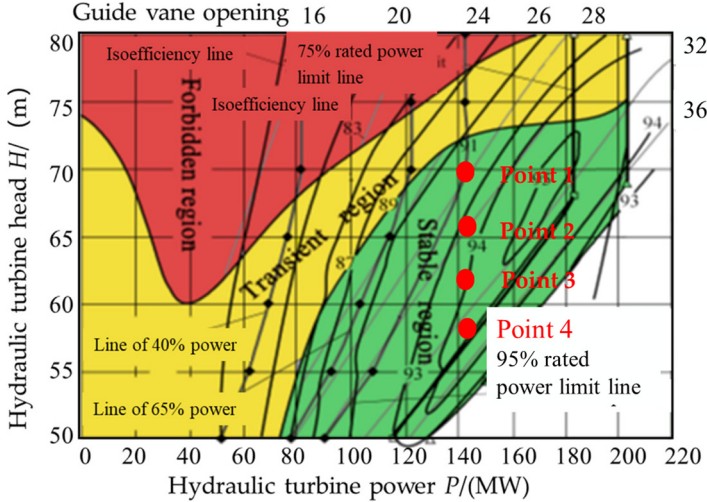

**Figure 4.** Comprehensive characteristic curve of water turbine.

According to the comprehensive characteristic curve, four water heads of 58 m, 62 m, 66 m, and 70 m were selected for numerical simulation. The guide vane opening was 24°, and the operating conditions selected for the calculation plan were in the stable region.

## 3. Numerical Simulation Reliability Verification

To verify the reliability of numerical simulation, external performance indicators and sediment erosion characteristics of the unit were used for validation. The operating performance and sediment erosion characteristics of the unit were tested, as shown in Figure 5, which is the experimental testing site. Table 1 shows the comparison of experimental and numerical results. The unit speed $n_{11}$ and unit flow rate $Q_{11}$ are defined as:

$$n_{11} = \frac{nD_{rn}}{\sqrt{H}} \tag{8}$$

$$Q_{11} = \frac{Q}{D_{rn}^2\sqrt{H}} \tag{9}$$

where $n$ is rotational speed, $H$ is head, $Q$ is flow rate, and $D_{rn}$ is runner diameter.

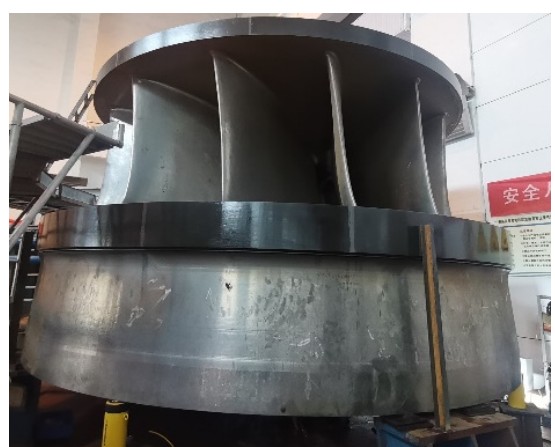

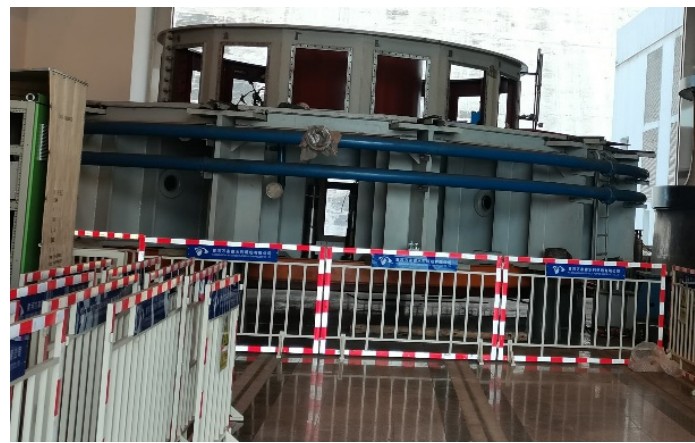

(**a**) Runner          (**b**) Top cover

**Figure 5.** Power station site photograph.

**Table 1.** Comparison of experimental and numerical results.

| Point | CFD Efficiency | Experimental Efficiency | $n_{11}$ | $Q_{11}$ |
|---|---|---|---|---|
| Point 1 | 94.05% | 94.51% | 77.85 | 0.7864 |
| Point 2 | 94.72% | 94.32% | 78.55 | 0.3678 |
| Point 3 | 94.25% | 94.24% | 91.21 | 0.3495 |

The numerical simulation predictions of the runner, movable guide vane, end clearance of the guide vane, top cover leakage ring, and bottom ring leakage ring are consistent with the actual sediment erosion situation of the unit, as shown in Figures 6 and 7. According to the sediment erosion condition of many years of operation, the actual erosion is consistent with the erosion damage position of the numerical simulation results, which indicates that the Tabakoff erosion model can accurately predict the erosion effect of the sediment-laden flow of the Francis turbine on the turbine components, which is in line with the engineering practice. Based on the above analysis, it can be concluded that the numerical simulation results are reliable.

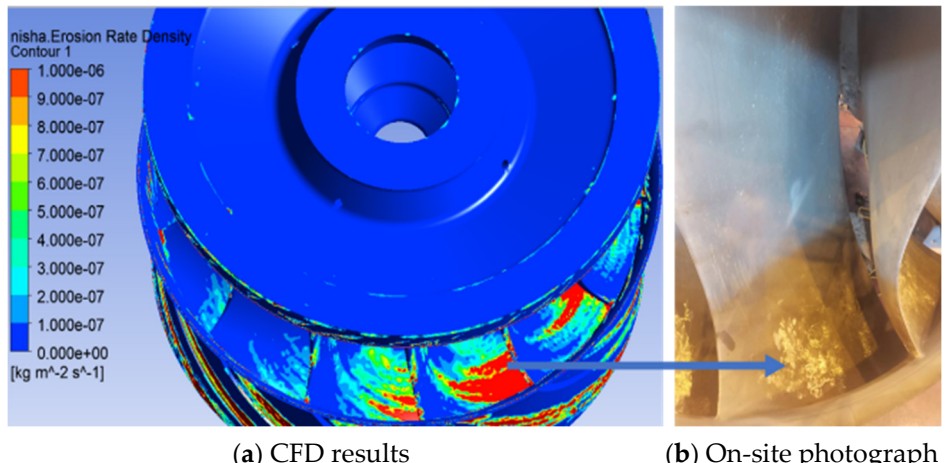

(**a**) CFD results       (**b**) On-site photograph

**Figure 6.** Comparison of sediment erosion of the runner obtained by CFD and real results on site.

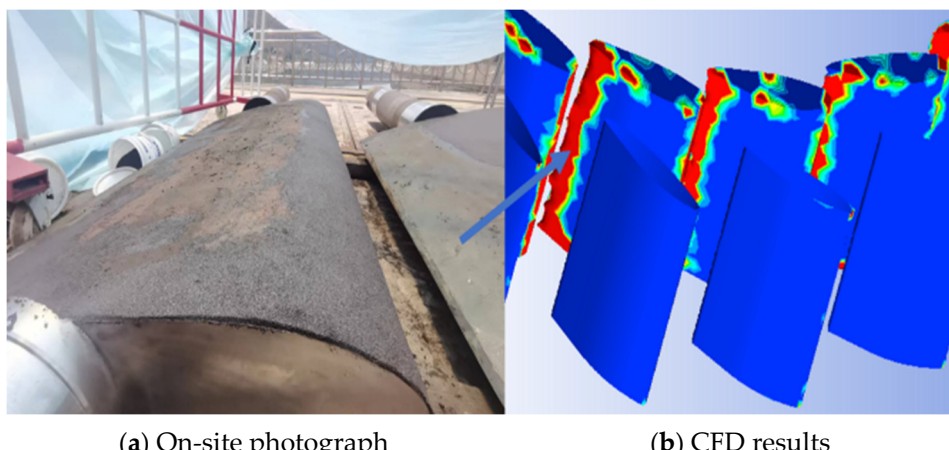

(**a**) On-site photograph       (**b**) CFD results

**Figure 7.** Comparison of sediment erosion of guide vane obtained by CFD and real results on site.

## 4. Analysis of Calculation Results

### 4.1. Flow Pattern in the Unit

Figure 8 shows the overall flow pattern distribution in the flow channels of different underwater units. The maximum velocities in the flow channel under different water heads are 25.68 m/s, 26.05 m/s, 28.17 m/s, and 29.07 m/s, respectively. This indicates that the higher the water head, the greater the flow velocity in the flow channel of the unit, and the higher the impact of the water flow on the unit wall. The flow pattern observation under different water heads is mainly shown in the draft tube. The higher the water head is, the more stable the flow pattern in the draft tube is, and the smaller the water head is, the worse the flow pattern in the draft tube is. There are some vortex backflows, which have a great impact on the particle movement in the flow channel. According to the existing research, particles will cause vortex sediment erosion on the wall when flowing at low velocity.

### 4.2. Sediment Erosion at the Guide Vane and Stay Vane

The sediment erosion situation at the guide vane and stay vane calculated using the Tabakoff model is shown in Figure 9. Due to the asymmetry of the circumferential flow field, it can be seen from Figure 9 that the degree and location of sediment erosion on each guide vane are not completely the same. However, the main areas of sediment erosion on the guide vane are concentrated in three positions, as shown in Figure 9. The vertical end face of the guide vane head is due to the impact of high-speed water flow carrying sediment on the front of the wall, resulting in wall erosion. There is significant sediment erosion between the end faces of the guide vanes. When the water heads are 66 m and 70 m,

the sediment erosion of the guide vanes is significantly more severe than when the water heads are 58 m and 62 m, indicating that sediment particles will cause severe sediment erosion on the guide vanes under high water heads.

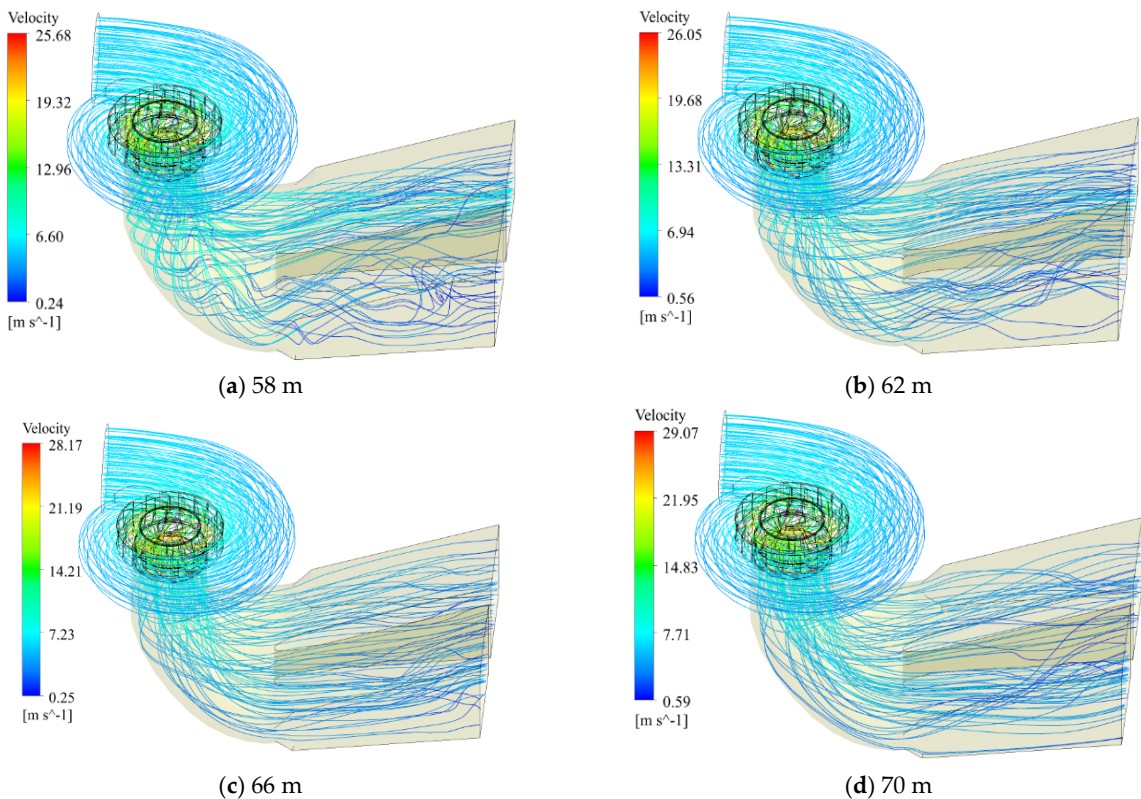

(**a**) 58 m                    (**b**) 62 m

(**c**) 66 m                    (**d**) 70 m

**Figure 8.** Flow pattern in the unit.

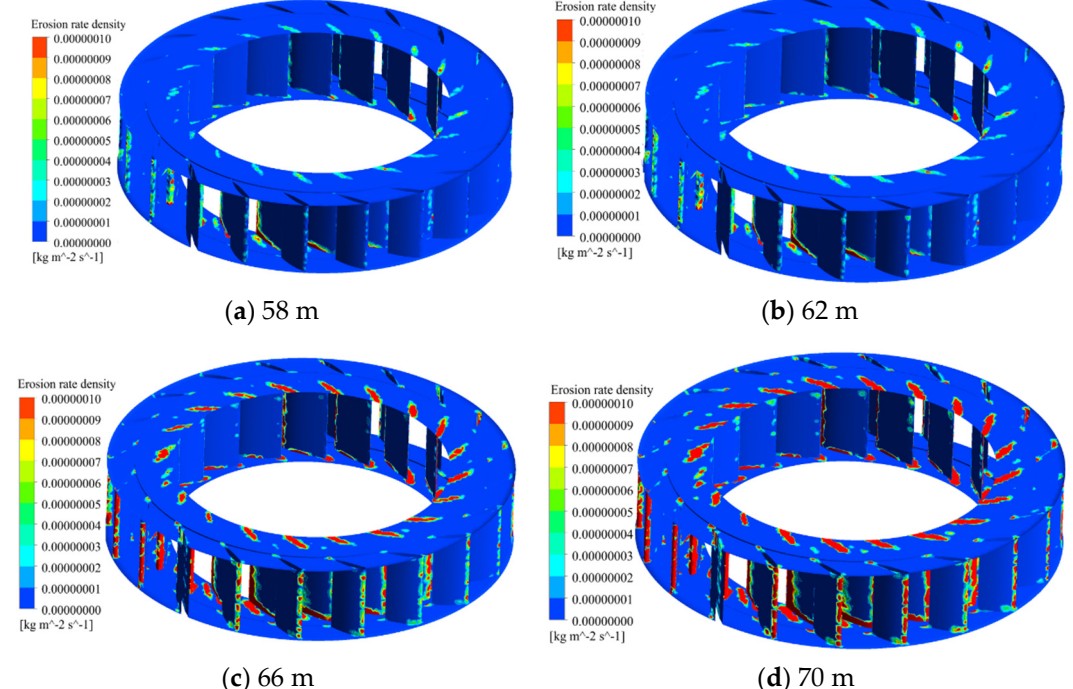

(**a**) 58 m                    (**b**) 62 m

(**c**) 66 m                    (**d**) 70 m

**Figure 9.** Sediment erosion at the guide vane and stay vane.

### 4.3. Runner Blade Area

As the core component of a hydraulic turbine, the state of the flow field around the runner is very important. Therefore, the velocity field and velocity vector around the runner are analyzed. The sediment erosion and tear of the upper crown and lower ring of the runner are shown in Figure 10. From the figure, it can be seen that there is only a small part of the upper crown surface, that is, there is slight sediment erosion at the connection between the inlet and outlet of the runner blade and the upper crown. The erosion area of the lower ring is the connection between the runner blade and the lower crown surface. From the perspective of various components of the entire hydraulic turbine, the sediment erosion and tear in the runner area is also the most severe. This is because sediment particles collide and rebound, resulting in wall erosion. On the suction surface of the runner inlet, the sediment velocity vector is parallel to the blade profile, and there is no collision contact between the sediment. Therefore, little erosion occurred on the suction surface of the blades. The location where erosion occurs is the same as the location where cavitation often occurs in hydraulic turbines, indicating that the flow pattern in these areas is easily disturbed and vortices appear, resulting in local high-speed flow. At the same time, the combined effect of erosion and cavitation further strengthens the destructive effect of erosion, forming a vicious cycle.

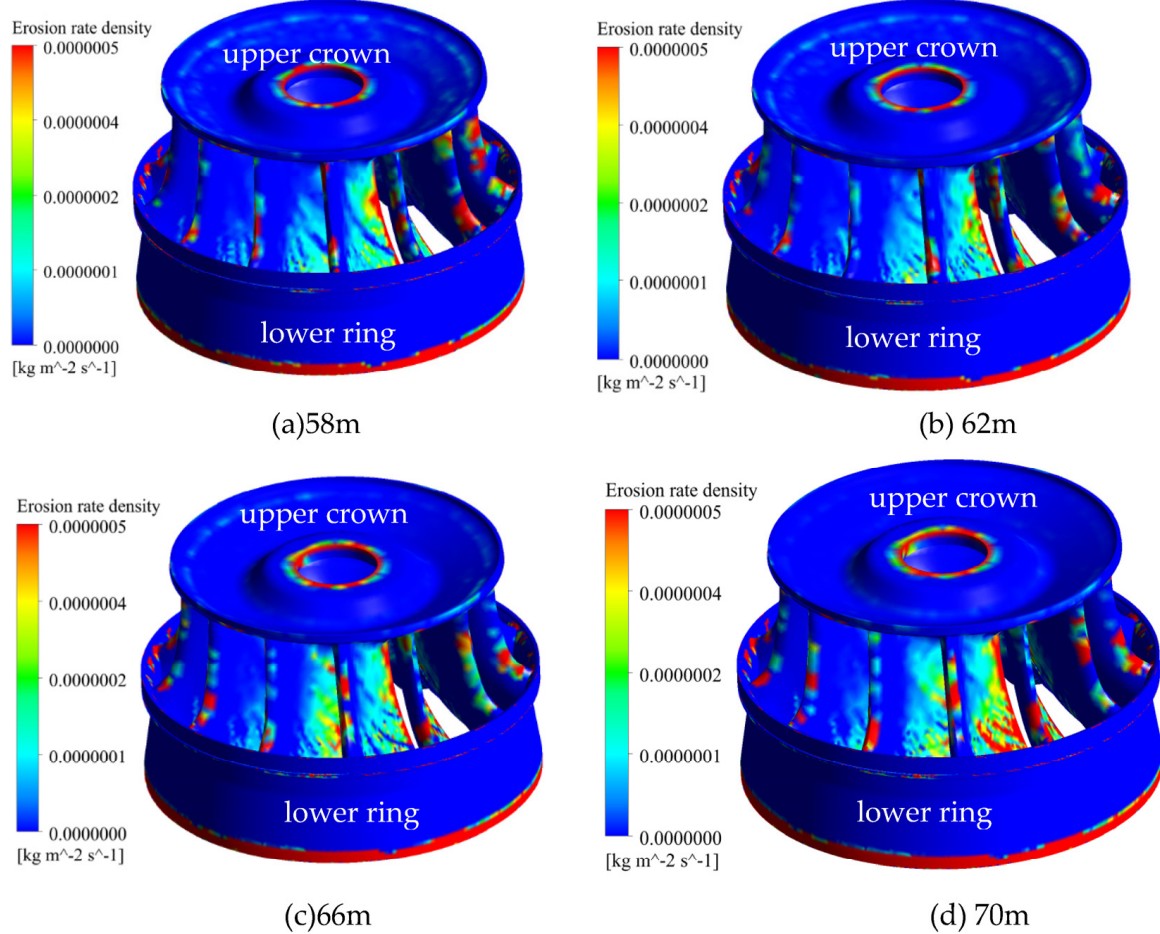

(a) 58m     (b) 62m

(c) 66m     (d) 70m

**Figure 10.** Sediment erosion at the runner.

### 4.4. Velocity Distribution at Runner

According to relevant research in hydraulic machinery, speed is the main cause of sediment erosion [17,18]. Analyzing the speed inside the runner under different water heads reveals the reasons for the differences in sediment erosion inside the runner under different water heads. Figure 11 shows the velocity distribution in the runner under

different water heads. Generally, the larger the velocity, the stronger the energy carrying sand, and the stronger the erosion effect on the wall. From the velocity field around the runner area in Figure 11, it can be inferred that the velocity distribution is similar along the circumference, and the larger areas of the velocity field are distributed at the outlet edge of the runner blades. Therefore, it can be inferred that the outlet edge of the runner blades is more prone to erosion. There is a high-speed zone at the lower ring position of the runner under different water heads, and the higher the water head, the higher the speed in the high-speed zone. The gap wall surface of the lower ring is also a high-speed zone, which is consistent with the sediment erosion distribution of the runner.

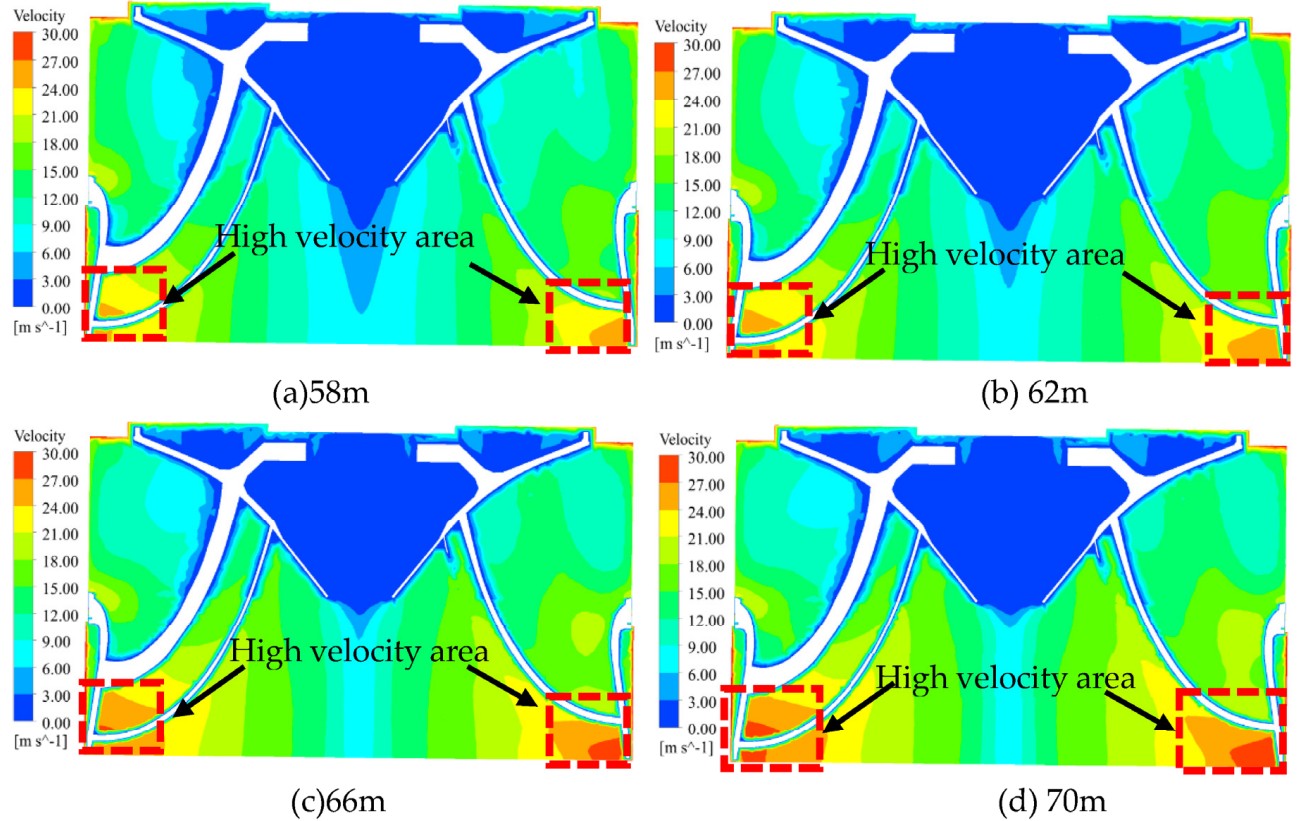

**Figure 11.** Velocity distribution at the runner.

According to the existing research results on friction in hydraulic machinery, it can be seen that the sediment erosion on the wall can be divided into impact sediment erosion and frictional sediment erosion, which are mainly determined by the impact angle of sediment particles [19]. The default impact angle of sediment particles set in the Tabakoff sediment erosion model is 25° [20], indicating that the impact angle of sediment particles when frictional sediment erosion occurs on the blade wall is below 25°. The impact speed mainly affects the degree of sediment erosion. The larger the impact speed of particles in the sediment erosion area, the more severe the sediment erosion. Figures 7, 9 and 10 show the particle's motion trajectory, sediment erosion degree, and impact speed. Based on the particle's motion characteristics and sediment erosion degree, it can be seen that the sediment erosion in the lower ring of the runner is mainly frictional sediment erosion. The high-speed zone at the lower end of the lower ring indicates that the higher the water head, the more severe the sediment erosion.

## 5. Conclusions

(1) The Tabakoff particle erosion model accurately predicts the erosion of sediment during the operation of the water turbine, which can provide guidance for the prevention of turbine erosion and the optimization design of the runner in the later stage.

(2) The main areas where erosion occurs are irregular boundary areas, including the area near the tongue of the volute, the unsmooth gaps or areas at the interface between surfaces, and the head of the guide vanes and blades directly impacted by sand-laden water flow.

(3) The unstable flow pattern inside the hydraulic turbine plays a decisive role in the severity of sediment erosion and tear. The area where the most severe sediment erosion and tear damage is calculated in this article is the blade area near the lower ring at the outlet of the runner. The curvature of the blade bending here is relatively large, resulting in high velocity and unstable flow phenomena. The optimization design of anti-sediment erosion and tear should focus on improving this area.

**Author Contributions:** Conceptualization, J.W.; methodology, J.W.; software, X.S.; validation, X.S.; formal analysis, X.S.; investigation, X.S.; resources, Z.W.; writing—original draft preparation, X.S.; writing—review and editing, visualization, Z.W.; supervision, R.T.; project administration, Z.W.; funding acquisition, H.W. All authors have read and agreed to the published version of the manuscript.

**Funding:** This work was supported by the Wanjiazhai Hydropower Station Hydro Generator Unit Stability Research Project (WJZ-ZB-2022-027).

**Data Availability Statement:** Not applicable.

**Conflicts of Interest:** The authors declare no conflict of interest.

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
