# Peer review of "Numerical Prediction of Erosion of Francis Turbine in Sediment-Laden Flow under Different Heads"

_processes, doi:10.3390/pr11092523_

Round 1

Reviewer 1 Report

The article solves the practical problem of erosion of water turbine blades using mathematical modeling using CFD methods. The results of mathematical modeling are clearly graphically marked and described in the conclusion. The achieved results in mathematical modelling are practically usable in technical practice when modifying water turbine blades.

The very good level of the article is partly reduced by some formal shortcomings (missing description of parameters H, P in Fig. 3, incomplete description of equations 8, 9 – what is Drn?, line 207 – literature 116?) and different level of graphic material (lower quality Fig. 3).

Despite the shortcomings mentioned, I recommend, after editing, publishing the article.

Author Response

Dear Editors and Reviewers:

Thank you for your letter and for the reviewers’ comments concerning our manuscript entitled “Numerical prediction of erosion of Francis turbine in sediment laden flow under different heads” (processes-2461114). Those comments are all valuable and very helpful for revising and improving our paper, as well as the important guiding significance to our researches. We have studied comments carefully and have made correction which we hope meet with approval. Revised portion are marked in the paper. The main corrections in the paper and the responds to the reviewer’s comments are as flowing:

Responds to reviewer #1: The very good level of the article is partly reduced by some formal shortcomings (missing description of parameters H, P in Fig. 3, incomplete description of equations 8, 9 – what is Drn?, line 207 – literature 116?) and different level of graphic material (lower quality Fig. 3).

Response: Thank you very much. I have made the necessary modifications and marked them in red in the paper.

Reviewer 2 Report

After reviewing the manuscript processes-2461114 titled “Numerical prediction of erosion of Francis turbine in sediment laden flow under different heads” by a group of authors, the following can be observed:

The manuscript is interesting and modern, but the manuscript itself and the presented idea cannot be considered original. The originality may only be reflected in the results presented for that particular case investigated, but it is not sufficient. Also, the novel contributions of these authors are not clearly emphasized.

The reviewer suggests that the authors include in their manuscript the citations of numerous other papers that deal with a similar topic, and which are not listed in the reference list. The authors should put effort into referring other relevant and older papers by significant worldwide research teams as well. This would show their quality and in-depth knowledge of the issues they deal with.

The authors must also pay special attention to their own reference list in order not to have this manuscript treated as an attempt at autoplagiarism. For example:

Song X, Zhou X, Song H, Deng J, Wang Z. Study on the Effect of the Guide Vane Opening on the Band Clearance Sediment Erosion in a Francis Turbine. Journal of Marine Science and Engineering. 2022; 10(10):1396. https://doi.org/10.3390/jmse10101396

Also, the reviewer would like to suggest the authors to correct chapter 2. (Materials and Methods) and explain some more details about the CFD accuracy and validation. The criterion for the choice of the appropriate mesh for numerical simulations is not completely clear. Hence, Grid Convergence Index (GCI) analysis is suggested to be done.

Discussions in the manuscript are poor and should be supported by a more profound and detailed analysis.

The manuscript should undergo a major revision in the form of some additions and clarifications.

Author Response

Dear Editors and Reviewers:

Thank you for your letter and for the reviewers’ comments concerning our manuscript entitled “Numerical prediction of erosion of Francis turbine in sediment laden flow under different heads” (processes-2461114). Those comments are all valuable and very helpful for revising and improving our paper, as well as the important guiding significance to our researches. We have studied comments carefully and have made correction which we hope meet with approval. Revised portion are marked in the paper. The main corrections in the paper and the responds to the reviewer’s comments are as flowing:

Comment 1: The manuscript is interesting and modern, but the manuscript itself and the presented idea cannot be considered original. The originality may only be reflected in the results presented for that particular case investigated, but it is not sufficient. Also, the novel contributions of these authors are not clearly emphasized.

Response: Thank you very much. I have made the necessary modifications and marked them in red in the paper.

Comment 2: The reviewer suggests that the authors include in their manuscript the citations of numerous other papers that deal with a similar topic, and which are not listed in the reference list. The authors should put effort into referring other relevant and older papers by significant worldwide research teams as well. This would show their quality and in-depth knowledge of the issues they deal with.

Response: Thank you very much. I have made the necessary modifications on the reference and marked them in red in the paper.

Comment 3: The authors must also pay special attention to their own reference list in order not to have this manuscript treated as an attempt at autoplagiarism. For example:

Song X, Zhou X, Song H, Deng J, Wang Z. Study on the Effect of the Guide Vane Opening on the Band Clearance Sediment Erosion in a Francis Turbine. Journal of Marine Science and Engineering. 2022; 10(10):1396. https://doi.org/10.3390/jmse10101396

 Response: Thank you very much. I have made the necessary modifications. And this paper is different form our previous paper.

Comment 4: Also, the reviewer would like to suggest the authors to correct chapter 2. (Materials and Methods) and explain some more details about the CFD accuracy and validation. The criterion for the choice of the appropriate mesh for numerical simulations is not completely clear. Hence, Grid Convergence Index (GCI) analysis is suggested to be done.

Response: Thank you very much. I have made the necessary modifications and marked them in red in the paper.

Comment 5: Discussions in the manuscript are poor and should be supported by a more profound and detailed analysis.

Response: Thank you very much. I have made the necessary modifications and added more discussions and marked them in red in the paper.

Comment 6: The manuscript should undergo a major revision in the form of some additions and clarifications.

Response: Thank you very much. I have made the necessary modifications.

Round 2

Reviewer 2 Report

After reviewing the revised manuscript processes-2461114-peer-review-v2 titled "Numerical prediction of erosion of Francis turbine in sediment laden flow under different heads" and the authors’ responses, the following can be concluded:

One of the authors stated several times: I made the necessary changes and marked them in red in the paper.

Although it is indicated that the entire abstract has been changed, the author has not made any changes, so Editors and Reviewers may be misled that something crucial has been done to the text. The previously mentioned also applies to the conclusion, so such an approach is inadmissible, and unfair on the part of the author (if the other co-authors saw the changes at all).

A descriptive listing of Grid independence verification, without any results and confirmation in the form of values in a table or diagram is not enough. The aforementioned supplemented discussion and references are insufficient, especially since the generally known facts are stated, and not a discussion of specific results, and because the authors did not consider very high-quality papers older than 15 years.

The authors should put effort into referring other relevant and older papers by significant worldwide research teams as well.

Therefore, from my point of view, the paper should again undergo a major revision.

Author Response

Thanks you again. I have made modifications based on your feedback, but the selection of literature was based on the research needs. We believe that each literature can provide assistance for this study, rather than being selected based on the publication date of the literature.